# Electrochemiluminescence Sensor Based on CTS-MoS_2_ and AB@CTS with Functionalized Luminol for Detection of Malathion Pesticide Residues

**DOI:** 10.3390/foods12234363

**Published:** 2023-12-03

**Authors:** Zhiping Yu, Chengqiang Li, Jiashuai Sun, Xia Sun, Guodong Hu

**Affiliations:** 1Shandong Key Laboratory of Biophysics, Dezhou University, Dezhou 253023, China; yuzhiping900414@aliyun.com; 2School of Agricultural Engineering and Food Science, Shandong University of Technology, No. 266 Xincun Xilu, Zibo 255049, China; jinclee527@gmail.com (C.L.); sunjiashuai97@163.com (J.S.)

**Keywords:** electrochemiluminescence, aptamer, CTS-MoS_2_, AB@CTS, PLL-luminol, nano gold

## Abstract

The accumulation of pesticide residues poses a significant threat to the health of people and the surrounding ecological systems. However, traditional methods are not only costly but require expertise in analysis. An electrochemiluminescence (ECL) aptasensor was developed using chitosan and molybdenum disulfide (CTS-MoS_2_), along with acetylene black (AB@CTS) for the rapid detection of malathion residues. Due to the weak interaction force, simple composite may lead to uneven dispersion; MoS_2_ and AB were dissolved in CTS solution, respectively, and utilized the biocompatibility of CTS to interact with each other on the electrode. The MoS_2_ nanosheets provided a large specific surface area, enhancing the utilization rate of catalytic materials, while AB exhibited excellent conductivity. Additionally, the dendritic polylysine (PLL) contained numerous amino groups to load abundant luminol to catalyze hydrogen peroxide (H_2_O_2_) and generate reactive oxygen species (ROS). The proposed ECL aptasensor obtained a low detection limit of 2.75 × 10^−3^ ng/mL (S/N = 3) with a good detection range from 1.0 × 10^−2^ ng/mL to 1.0 × 10^3^ ng/mL, demonstrating excellent specificity, repeatability, and stability. Moreover, the ECL aptasensor was successfully applied for detecting malathion pesticide residues in authentic samples with recovery rates ranging from 94.21% to 99.63% (RSD < 2.52%). This work offers valuable insights for advancing ECL sensor technology in future applications.

## 1. Introduction

In agricultural production, organophosphorus pesticides are utilized as organic compounds to control plant diseases, insects and pests, and so on. This kind of pesticide encompasses a wide range that exhibit high extensive usage. Moreover, they possess the advantage of easy decomposition and generally do not accumulate in human and livestock bodies. The compound is of the utmost importance in the field of pesticides and has become more and more important in agricultural production [1]. Excessive utilization of malathion, an extensively employed insecticide, has resulted in the accumulation of its residues within the environment. Consequently, the presence of malathion residues presents potential hazards to the well-being of humans and the surrounding ecological systems [2].

Currently, various conventional techniques are available for identifying organo-phosphorus pesticides. These techniques encompass gas chromatography (GC) [3,4,5], high-performance liquid chromatography (HPLC) [6,7], gas chromatography–mass spectrometry (GC-MS) [8,9], infrared spectroscopy [10], and the enzyme-linked immunosorbent test (ELISA) [11,12,13]. Even though these conventional methods offer high precision and stability, they often entail significant expenses and time-consuming sample preparation, in addition to demanding expert analytical skills that make these methods unsuitable for field detection and rapid detection [14]. Therefore, researchers have tried to use various rapid detection methods in analysis to detect pesticide residues in samples [15]. The biosensor, as a rapid detection technology, exhibits advantages of low cost, simple operation, and fast detection in identifying pesticide residues. Consequently, it has garnered significant interest from numerous scientific researchers who are engaged in extensive research and exploration in this field. The existing expedited detection techniques encompass fluorescent biosensors [16], colorimetric biosensors [17,18], electrochemical biosensors [19], and other methods.

The electrochemiluminescence (ECL) biosensor aims to accurately and sensitively detect the concentration of the target substance by analyzing the relationship between its concentration and the signal intensity of the electrochemiluminescence sensor [20,21,22]. With its superior sensitivity, minimal background signal, and swift response, this biosensor provides reliable detection capabilities; this method is widely used in pesticide detection, immunoassay, and the diagnosis of cancer [23,24]. In recent years, the field of nanotechnology has witnessed significant advancements in the development of pesticide residue detection methods. These methods are constructed using various nanomaterials, leading to improved sensitivity and stability in rapid pesticide residue detection and analysis technology [25].

Aptamers are well known for their small size, non-immunogenicity, superior stability, and distinctive three-dimensional (3D) structures. Due to their specific recognition capabilities, demonstrating high sensitivity and maintaining stability, aptamer-based electrochemical biosensors have gained significant traction in research and practical applications. To enhance the sensitivity of the electrochemical aptasensor, various materials have been developed and synthesized to enable modifications on the working electrode.

Luminol is a molecule that exhibits chemo-fluorescence properties. This molecule efficiently transforms into an excited amino-phthalic acid when hydrogen peroxide molecules are present, leading to the generation of intense fluorescence. Hydrogen peroxide, derived from diverse biological oxidation reactions, presents a convenient linkage between these reactions and photodetection, making it convenient to connect these reactions to photodetection through the use of luminol. Among the electrochemiluminescence sensors, luminol demonstrates outstanding performance in terms of stability, luminescence rate, and cost-effectiveness, making it widely utilized [26].

Nano gold (AuNPs) are utilized in ECL sensors, which possess excellent electronic properties, as well as the ability to enhance signals. Moreover, they not only effectively improve the ECL signal of luminol reagent but establish a strong connection with the aptamer through covalent bonds [27,28]. 

Dendrimers are monodisperse macromolecules with regular, highly branched three-dimensional structures. The surface of dendritic polylysine (PLL) exhibits numerous functional groups, excellent chemical stability, and a highly symmetric geometric structure [29]. The secure immobilization of nanoparticles on the PLL surface is facilitated by these characteristics.

Chitosan (CTS) is a type of polysaccharide obtained through the deacetylation of chitin. Due to its exceptional film-forming ability, adsorption, biocompatibility, and porous structure, it is widely used in the preparation of electrochemically modified electrodes [30,31]. Moreover, the CTS molecular chain contains -NH_2_ and -OH groups that exhibit strong chelation ability for heavy metal ions.

The two-dimensional (2D) layered nanomaterial is an emerging class of nanomaterials. Molybdenum disulfide (MoS_2_), as a representative of 2D nanomaterials, has been widely used in the field of electrochemistry due to its large surface area, high electron mobility, photoluminescence, abundant active sites, and so on. MoS_2_ presents various benefits due to its modifiable void structure, enhancing electron transfer and generating synergistic outcomes [32,33]. Moreover, the increased specific surface area of composites containing MoS_2_ allows for the incorporation of additional active probes and domains. This facilitates simple separation by sub-binding with organisms, leading to a considerable amplification of the electrochemical signal. These features establish firm groundwork for sensor construction [34,35,36].

Acetylene black (AB) is a unique form of carbon black typically produced through the compression and combustion of acetylene in an oxygen-rich environment. AB exhibits a significant specific surface area, excellent conductivity, and impressive adsorption capabilities, making it a highly sought-after material in the realm of electrochemistry [37,38,39].

In this research, we utilized CTS-MoS_2_ and AB@CTS to modify the electrode. Since the conductivity of single MoS_2_ is not ideal, we improved it by combining it with acetylene carbon black, which enhanced the conductivity of the nanomaterials. MoS_2_, as the first layer of the electrode, provided a large number of active sites, improving the utilization rate of catalytic materials due to its large specific surface area, which bridged the electrodes and nanomaterials. Additionally, AB, with its excellent electrical conductivity, improved the electron transfer rate, resulting in the amplification of the ECL signal. The system also included dendritic polylysine (PLL), which carried numerous amino groups for loading abundant luminol to catalyze hydrogen peroxide (H_2_O_2_) and generate reactive oxygen species (ROS), thereby enhancing the ECL signal. The aptamer was immobilized to the electrode surface using the Au-S bond with AuNPs. These prepared composites were combined to construct an ECL adaptation sensor that exhibited high sensitivity and strong specificity for detecting malathion residues.

## 2. Materials and Methods

### 2.1. Reagents and Materials

MoS_2_ was purchased from Merck Chemicals (Shanghai) Co., Ltd., Shanghai, China. Acetylene black (AB) was purchased from Tianjin Huayuan Chemical Technology Co., Ltd., Tianjin, China. Dendritic polylysine (PLL) (molecular weight: 20,000~40,000 and ammonic number: 270~540) was obtained from Beijing Ruida Henghui Technology Development Co., Ltd., Beijing, China. Gold chloride trihydrate, CTS, and luminol were obtained from Shanghai Aladdin Biochemical Technology Co., Ltd., Shanghai, China. Tri-Sodium citrate was obtained from Sinopharm Chemical Reagent Co., Ltd., Shanghai, China. The sequence of the DNA aptamer (Apt) was 5′-SH-ATCCGTCACACCTGCTCTTATAC ACAATTGTTTTTCTCTTAACTTCTTGACTGCTGGTGTTGGCTCCCGTAT-3′. The aptamer was diluted to 100 μM using tris-EDTA buffer solution. Bovine serum albumin (BSA), aptamers, and tris-EDTA buffer solution were purchased from Sangon Biotech (Shanghai) Co., Ltd., Shanghai, China. The phorate, profenofos, acetamiprid, carbofuran, and malathion were obtained from Beijing Zhongke Quality Inspection Biotechnology Co., Ltd., Beijing, China.

### 2.2. Apparatus

The detection of ECL was conducted by utilizing an ECL analyzer model MPI-A (Xi’an Remax Electronic Science & Technology Co., Ltd., Xi’an, China). To generate ultrapure water, the LS MK_2_ ultrapure water purification system (PALL Co., Ltd., Shanghai, China) was employed. During the detection process, a three-electrode system was utilized. The system consisted of the reference electrode (RE) Ag/AgCl, the counter electrode (CE), the platinum electrode (PE), and the working electrode which was a glassy carbon electrode (GCE). 

### 2.3. Preparation of CTS-MoS_2_

The preparation of 0.1% CTS solution was as follows: 1 mL glacial acetic acid was mixed into 99 mL ultrapure water and 0.1 g CTS powder was added. This was stirred for more than 3 h to completely dissolve CTS at room temperature. A total of 15 mg MoS_2_ was weighed and added into 10 mL of CTS solution, which underwent ultrasound for 12 h in ice bath conditions. This was then stirred using a magnetic stirrer for 2 h to form a 1.5 mg/mL CTS-MoS_2_ solution. It was stored in a refrigerator at 4 °C.

### 2.4. Preparation of AB@CTS

To prepare the AB@CTS solution, the initial step was that 10 mL of a 0.1% CTS solution was placed into a beaker. Then, 10 mg of AB was added and the mixture was stirred for 10 min. After that, the mixture was subjected to ultrasound for 6 h in an ice bath. Finally, the solution was continually stirred using a magnetic mixer for 2 h at room temperature to obtain a concentration of 1 mg/mL AB@CTS and the solution was stored at 4 °C.

### 2.5. Preparation of AuNPs

The experimental glassware underwent immersion in aqua regia for a duration exceeding 12 h. Subsequently, comprehensive washing and thorough drying procedures were performed. The initial step involved adding 1 g of trisodium citrate into 100 mL of ultrapure water and meticulously mixing it. After this, 2.5 mL of the trisodium citrate solution was added into 100 mL of ultrapure water. The resulting blend was then subjected to heating with an induction cooker on a high heat, achieving boiling, and was continuously stirred for two minutes. Following this, using a medium heat on the induction cooker, dropwise additions of 1 milliliter of a 1% chloroauric acid solution were meticulously executed with the condition of vigorous stirring. Subsequent to the addition, the solution was continuously stirred and heated for a duration of 15 min until it transformed into a visually striking wine-red color. At this juncture, heating was terminated, but stirring persisted for an additional 30 min to facilitate the production of AuNPs. Ultimately, the solution was subjected to centrifugation, followed by three subsequent steps of washing with ultrapure water, prior to adjusting its volume to 100 mL and storing it at 4 °C.

### 2.6. Preparation of PLL-Luminol

A total of 1 mg of PLL was added to 5 mL of ultrapure water and it was thoroughly stirred until it was well mixed. Subsequently, 2 mL of luminol solution (0.01 M) was added into it. Then, it was stirred for 2 h at room temperature in the dark. The PLL-luminol nanocomposite was obtained and stored in a dark container at 4 °C in a refrigerator.

### 2.7. Processing of the Actual Samples

Initially, the vegetables underwent initial cleansing and a subsequent drying process. Next, they were meticulously sliced into minuscule fragments of an approximate size of 1–2 mm. Then, 2 g of the vegetable pieces was measured and carefully deposited into designated centrifuge tubes. Following this, 2 mL of different concentrations of organophosphorus pesticide was sprayed into centrifuge tubes. The tubes were kept at room temperature for 24 h. Following this, 0.2 mL of acetone and 1.8 mL of 0.02 M PBS (PH = 8.0) buffer solution were introduced. Finally, the solution was mixed and underwent ultrasound for 1 h and was then centrifuged at 10,000 rpm for 10 min. The supernatant measurements were taken.

### 2.8. Fabrication of ECL Aptasensor

The fabrication process of the aptasensor, as depicted in Figure 1, demonstrated that the GCE needed to undergo a series of treatments to achieve optimal ECL. To achieve the desired chemical modification, nanomaterials were uniformly added to the electrode surface using the droplet application technique. After the solvent had evaporated, the coating film became bound to the electrode surface, thereby achieving the desired chemical modification. Initially, the GCE was polished using 0.05 μm alumina powder, resulting in a reflective and smooth surface. Subsequently, ultrasonic treatment was carried out in both water and ethanol. To begin, a coating of 5 μL CTS-MoS_2_ solution was applied onto the electrode surface, followed by a natural drying step at room temperature. Next, 5 μL AB@CTS solution was deposited onto the GCE surface and natural drying was performed as well. Afterward, 5 μL PLL-luminol solution was introduced and dried on the GCE surface. To enhance the luminous effect and immobilize malathion aptamers, 5 μL of AuNPs solution was applied onto the GCE surface and dried accordingly. Following this, a solution of thiol-modified aptamer (100 nM) was dropped onto the GCE in a volume of 5 μL. To prevent non-specific binding, 5 μL of BSA (0.5%) was then added and incubated for 30 min to block any unspecific sites. The process of aptasensor fabrication and the detection principle for MAL are illustrated in Figure 1.

## 3. Results

### 3.1. Characterization of Prepared Nanomaterials

To gain a more comprehensive overview of nanomaterial distribution on the assembled electrode’s surface, a SEM test was performed. It revealed a uniform dispersal of nanomaterials across the electrode surface (Figure 2A). The CTS-MoS_2_ was characterized using SEM and EDS, while the AuNPs were analyzed using TEM and UV-vis spectroscopy analysis. It can be seen that numerous AuNPs exhibited with a particle size around 15 nm and regular circular shapes, without any signs of aggregation, indicating successful preparation (Figure 2B,C). Additionally, the maximum UV absorption peak of the AuNPs colloidal solution was about 521.5 nm, further proving that it has a small particle size (Figure 2D). The original MoS_2_ has a massive structure with a thick interlayer. However, by subjecting it to high-frequency ultrasound waves, the interlayer motion energy was provided to facilitate the chelation ability of CTS, resulting in the effective stripping off MoS_2_ and the prevention of its agglomeration. Notably, this clearly demonstrates that the interlayer structure of MoS_2_ was transformed into a thinner nanosheet structure (Figure 2E). As shown in Figure 2F, this was the isothermal adsorption curve of MoS_2_. Through the test, the BET surface area of MoS_2_ was 3.1173 m^2^/g. Additionally, X-ray energy spectroscopy analysis proved the existence of Mo and S as two elements (Figure 2G–I).

### 3.2. Electrochemical Behavior of the Aptasensor

Cyclic voltammetry (CV) and electrochemical impedance spectroscopy (EIS) were employed to analyze the modified electrodes with varying materials. In Figure 3A, it can be observed that in comparison to the unmodified glassy carbon electrode (GCE), one or more introductions of CTS-MoS_2_, AB@CTS, or AuNPs onto GCE showed a higher current peak, indicating their excellent electrical conductivity. When PLL-luminol was modified onto the GCE, the current signal gradually decreased due to the reduction in aptasensor conductivity caused by luminol, which subsequently hindered the movement of free electrons and resulted in a decrease in current signal. Furthermore, upon modification of the electrode with BSA, the hindrance of free movement of redox probes within the reaction solution caused a notable decrease in the current peak. This decrease was additional and significant in nature.

We present the results of the electrochemical impedance spectroscopy (EIS) analysis for the characterization of the aptasensor assembly in Figure 3B. The bare electrode demonstrated a maximum-size semicircle, whereas the introduction of CTS-MoS_2_, AB@CTS or AuNPs modified GCEs resulted in a smaller semicircle. This can be attributed to the remarkable enhancement in electron transfer facilitated by these three materials. On the other hand, when PLL-luminol was employed as part of the modification, the resistance gradually increased. This can be attributed to the reduced conductivity of luminol, which acted as a hindrance for the flow of free electrons. Furthermore, upon modification with Apt and BSA on the electrode surface, the resistance further increased as both materials exhibited poor conductivity, thereby impeding electron transmission. The electrochemical impedance spectroscopy (EIS) results were consistent with those obtained from cyclic voltammetry (CV). These findings collectively demonstrate successful assembly of the aptasensor.

The electrochemical active surface area (ECSA) of an electrocatalyst composite provides valuable information about the electrochemically active molecules present. In this study, ECSA values were determined using double-layer capacitance (Cdl) in non-faradaic cycling voltammograms (CVs) at different scan rates, as shown in Figure 3C. The relationship between current density and scan rate was plotted and fitted with a linear equation, as seen in Figure 3D. The slope of this equation corresponded to a Cdl value of 0.005 mF/cm^2^. The ECSA of the electrode was then calculated using Equation (1), which is standard. Additionally, the specific capacitance Cs for metal surfaces was determined to be 35 μF·cm^−2^. A is the surface area of the electrode, which was 7.1 mm^2^. Based on these calculations, the ECSA of the electrode was found to be 101.46 mm^2^.
(1)$$ECSA=\frac{Cdl}{Cs}\;\times \;A$$

### 3.3. ECL Characterization of Different Modified Electrodes

To authenticate the practicability of the contrived aptasensor, the ECL parameters were configured with a scan rate of 0.1 V/s. The voltage utilized for the quantification of luminescence intensity was set at 700 V in the photomultiplier tube. Using 0.2 M PBS (PH = 8) containing 6 nM H_2_O_2_, we conducted a study to compare the ECL responses of various modified electrodes. Figure 4 presents the ECL responses of the different modified electrodes. While the modification of GCE with PLL-luminol displayed a relatively weak ECL signal, it was still stronger in comparison to the electrode that was solely modified with luminol. This is because the dendritic polylysine (PLL) in the system contained numerous amino groups, enabling the loading of abundant luminol to catalyze hydrogen peroxide (H_2_O_2_) and generate reactive oxygen species (ROS), thereby enhancing the ECL signal. Nevertheless, a remarkable ECL signal was detected following the introduction of CTS-MoS_2_, AB@CTS, or AuNPs onto the luminol-modified electrode. This observation strongly suggests that the conductivity and surface area of these three nanomaterials were exceptional. The ECL signal exhibited enhanced intensity upon introducing AB@CTS onto the GCE surface, owing to the remarkable catalytic activity of AuNPs. The presence of aptamers on the GCE resulted in a diminished ECL intensity, attributed to their disruptive effect on the ECL process. Furthermore, when GCE was modified with BSA, the ECL intensity was further decreased due to its macromolecular nature and impeding process.

### 3.4. Experiments of Optimization

In order to achieve the best results in analytical performance, several experimental parameters were fine-tuned. These parameters included optimizing the concentration of the aptamer, adjusting the pH level, controlling the concentration of H_2_O_2_, and ensuring the appropriate incubation time between the aptamer and malathion. In order to find the optimal dosage of immobilized aptamer, various concentrations of the aptamer were fixed onto the electrode surface. As depicted in Figure 5A, the biosensor’s ECL responses exhibited a steady increase, directly correlating with the concentration of the aptamer. Once the aptamer concentration reached 100 nM, the intensity of the ECL peaked, indicating that the aptamer on the GCE surface had reached a state of saturation. Based on this, it can be concluded that 100 nM is the ideal aptamer concentration for subsequent experimentation. 

The luminol-based ECL system demonstrated superior luminous properties under weak alkaline conditions as opposed to neutral or acidic environments. Consequently, it becomes imperative to optimize the pH of the reaction solution. As delineated in Figure 5B, the ECL intensity of the aptasensor exhibited a progressive enhancement with an escalation in pH. However, the pinnacle of the ECL intensity was observed at a pH of 8.0. Therefore, it was established that the optimum pH value for this experiment was determined to be 8.0.

To optimize the performance of the sensor, we focused on optimizing the incubation time for pesticides and aptamers. The amount of binding between the aptamer and the pesticides was influenced by the incubation time, which in turn affected the luminescence performance of the sensor. In Figure 5C, it can be observed that the luminescence intensity of the sensor gradually increased with longer incubation times. When the incubation time reached 40 min, the luminescence intensity stabilized, indicating that the pesticide and aptamer complex on the sensor’s surface had reached saturation. A further increase in incubation time did not result in any change in luminescence intensity. Therefore, we determined that 40 min was the optimal incubation time for achieving the desired interaction between the aptamer and the pesticide.

Figure 5D illustrates a notable enhancement in ECL intensity as the concentration of H_2_O_2_ in the bottom fluid significantly increased. Upon reaching a concentration of 6 mM, the ECL intensity reached its maximum level and was constantly maintained thereafter. This indicates that the electrochemiluminescence reaction reached its peak when the concentration of H_2_O_2_ was 6 mM. Therefore, the optimal concentration of H_2_O_2_ is 6 mM. 

### 3.5. Performance of ECL Aptasensor

The detection performance of the proposed aptasensor was assessed by obtaining and analyzing the ECL data at different concentrations of malathion. As shown in Figure 6, the proposed sensor showcased a remarkable decline in ECL intensity as the pesticide concentration ascended from 10^−2^ ng/mL to 1 μg/mL. This observation revealed an exceptional correlation between the two variables, indicative of a linear relationship. The linear expression y = 5920.41 − 659.23LogC_malathion_ accurately depicts this relationship, with a substantial correlation coefficient (R^2^) of 0.995. Here, y denotes the ECL intensity, while LogC represents the logarithm of malathion concentration. Through calculations, we determined the detection limit (LOD) as per the following procedure.
(2)$$LOD=\frac{3\times SD}{K}$$

The standard deviation (SD) indicates the standard deviation of the signal from the blank standard specimen, while K represents the slope of the calibration curve of the aptasensor. By considering a signal-to-noise ratio S/N = 3, the limit of detection (LOD) was determined to be 0.00275 ng/mL.

As MoS_2_ has a modifiable void structure, electron transfer can be enhanced and synergistic outcomes can be generated. In addition, AB has excellent electrical conductivity, further promoting electrochemiluminescence. The AuNPs amplified the ECL signal through their catalytic effect. Furthermore, the inclusion of dendritic polylysine (PLL) in the system offered numerous amino groups, which enabled the loading of abundant luminol for the catalysis of H_2_O_2_ and the generation of reactive oxygen species (ROS), ultimately resulting in enhanced ECL signal detection. In addition, the aptamer can bind to the corresponding ligand with high affinity and strong specificity. Therefore, in comparison to alternative detection techniques, this method exhibited a lower limit of detection and a broader linear detection range, as depicted in Table 1.

### 3.6. Specificity, Repeatability, and Stability of the Aptasensor

Our investigation focused on evaluating the performance of the aptasensor in terms of its specificity, repeatability, and stability. The specificity stands out as a crucial performance indicator. To assess the aptasensor effectively, we selected four common interference species typically found in vegetable production. These species include phorate, profenofos, acetamiprid, and carbofuran. Figure 7A illustrates that the mentioned pesticides had a negligible impact on the luminescence intensity of the electrodes. However, in the presence of malathion, a significant decrease in ECL intensity was observed. Consequently, these results strongly indicate that the aptasensor exhibits exceptional specificity, specifically toward malathion.

Stability was a crucial characteristic of the aptasensor. In Figure 7B, the ECL responses are documented at regular intervals of 4 days throughout a duration of 16 days. The ECL signals were reduced by approximately 1.68%, 3.07%, 5.81%, and 7.79% in comparison to the initial peak intensity after of 4, 8, 12, and 16 days, respectively, thereby further validating the stability of the aptasensor.

Additionally, the repeatability was also a crucial property. By analyzing the ECL response from ten distinct electrodes independently designed under identical conditions, we illustrated that there were no discernible discrepancies in the response signals of every ECL aptasensor, as seen in Figure 7C. These electrodes exhibited analogous responses, displaying a relative standard deviation (RSD) of 1.18%, thereby indicating excellent repeatability for the fabricated aptasensor.

### 3.7. Real Sample Analysis in Vegetables

To assess the analytical capabilities of aptasensors in practical scenarios, we employed the standard addition technique to examine the presence of MAL residues in vegetables. Our findings revealed recovery rates spanning from 94.21% to 99.63% accompanied by an RSD varying between 1.38% and 2.52%. These results are summarized in Table 2, demonstrating that the ECL aptasensor holds promising prospects for MAL detection in vegetable samples.

## 4. Conclusions

A novel method was developed in this research study to detect malathion residues in vegetables using an electrochemiluminescence (ECL) aptasensor. The aptasensor design incorporated several key components, including PLL-luminol, CTS-MoS_2_, AB@CTS, AuNPs, and aptamer. The integration of CTS-MoS_2_, known for its large surface area, provided numerous active sites that enhanced the utilization rate of catalytic materials and facilitated bridging between the electrodes and nanomaterials. Additionally, the excellent electrical conductivity of AB@CTS further improved the ECL signals of the aptasensor. However, due to weak interaction forces, the simple composite resulted in uneven dispersion, affecting the overall stability of the material. To overcome this, both MoS_2_ and AB were dissolved in CTS solution, respectively, and utilized the biocompatibility of CTS to interact with each other on the electrode, enhancing the luminous intensity of the ECL aptasensor. The presence of AuNPs not only amplified the ECL signal through their catalytic effect, but also facilitated the modification of aptamers on the electrodes through Au–S bonds. Furthermore, the inclusion of dendritic polylysine (PLL) in the system offered numerous amino groups, which enabled the loading of abundant luminol for the catalysis of hydrogen peroxide (H_2_O_2_) and the generation of reactive oxygen species (ROS), ultimately resulting in enhanced ECL signal detection. The detection of malathion in real samples was effectively achieved by the aptasensor proposed in this study, which exhibited remarkable stability, superior sensitivity, and wide applicability. Following the optimization of experimental conditions, the ECL aptasensor displayed a linear detection range spanning from 1.0 × 10^−2^ ng/mL to 1.0 × 10^3^ ng/mL. In contrast to previously employed detection techniques, this aptasensor exhibited superior attributes, including a lower detection limit (2.75 × 10^−3^ ng/mL) (S/N = 3) and an expanded detection range. Finally, the ECL aptasensor proposed in this study was successfully employed for the detection of malathion pesticide residues in real samples, with recovery rates ranging from 94.21% to 99.63% and RSD being within the range of 1.38% to 2.52%. This study has provided valuable insights for the development of ECL sensors in the future: (1) The sensitivity and selectivity of ECL detection techniques can be enhanced through an improvement in substrate materials and modification of electrode surfaces. This can further expand the application of ECL sensor technology in pesticide residue detection and facilitate the development of intelligent and integrated portable ECL analysis assemblies. These advancements would enable real-time monitoring of pesticide residues. (2) It is important to explore new and highly efficient ECL systems and utilize them for the development of flexible luminescence devices.

## Figures and Tables

**Figure 1 foods-12-04363-f001:**
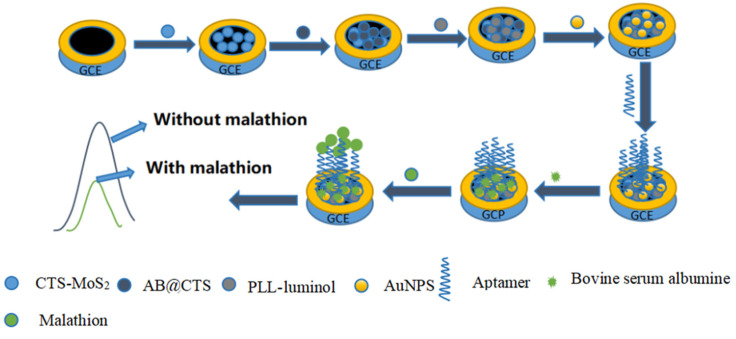
The schematic diagram outlines the process of creating the ECL aptamer sensor.

**Figure 2 foods-12-04363-f002:**
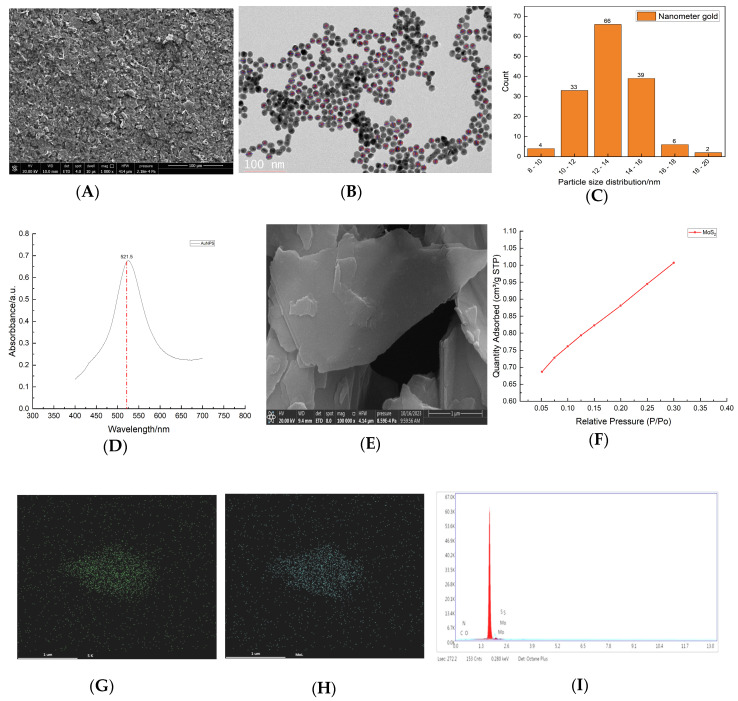
Characterization of prepared nanomaterials. (**A**) The SEM of the assembled electrode, (**B**) the TEM of AuNPs, (**C**) histogram of AuNPs particle size distribution, (**D**) UV–vis spectroscopy analysis of AuNPs, (**E**) the SEM of the stripped MoS_2_, (**F**) isothermal adsorption curve of MoS_2_, (**G**) the EDS element mapping graph of Mo, (**H**) the EDS element mapping graph of S, and (**I**) the EDS energy spectra of CTS-MoS_2_.

**Figure 3 foods-12-04363-f003:**
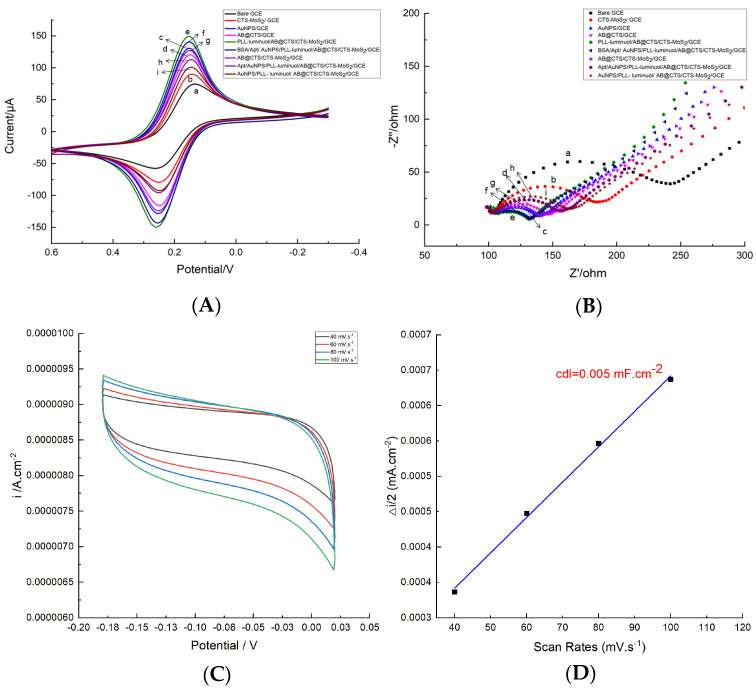
CV (**A**) and EIS (**B**) of different modified GCEs: (a) bare GCE, (b) CTS-MoS_2_/GCE, (c) AB@CTS/CTS-MoS_2_/GCE, (d) PLL-luminol/AB@CTS/CTS-MoS_2_/GCE, (e) AuNPs/PLL-luminol/AB@CTS/CTS-MoS_2_/GCE, (f) Apt/AuNPs/PLL-luminol/AB@CTS/CTS-MoS_2_/GCE, (g) BSA/Apt/ AuNPs/PLL-luminol/AB@CTS/CTS-MoS_2_/GCE, (h) AB@CTS/GCE, and (i) AuNPs/GCE. (**C**) CV curves for electrocatalyst with overpotential window of −0.18~0.02 V and (**D**) calculated ECSA value.

**Figure 4 foods-12-04363-f004:**
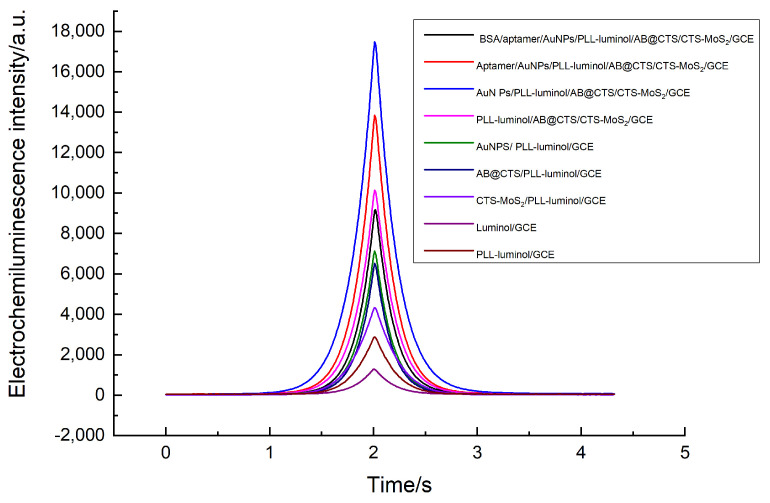
Electrochemiluminescence curves of different modified electrodes: BSA/aptamer/AuNPs/PLL-luminol/AB@CTS/CTS-MoS_2_/GCE, aptamer/AuNPs/PLL-luminol/AB@CTS/CTS-MoS_2_/GCE, Au NPS/PLL-luminol/AB@CTS/CTS-MoS_2_/GCE, PLL-luminol/AB@CTS/CTS-MoS_2_/GCE, AuNPs/PLL-luminol/GCE, AB@CTS/PLL-luminol/GCE, CTS-MoS_2_/PLL-luminol/GCE, luminol/GCE, and PLL-luminol/GCE.

**Figure 5 foods-12-04363-f005:**
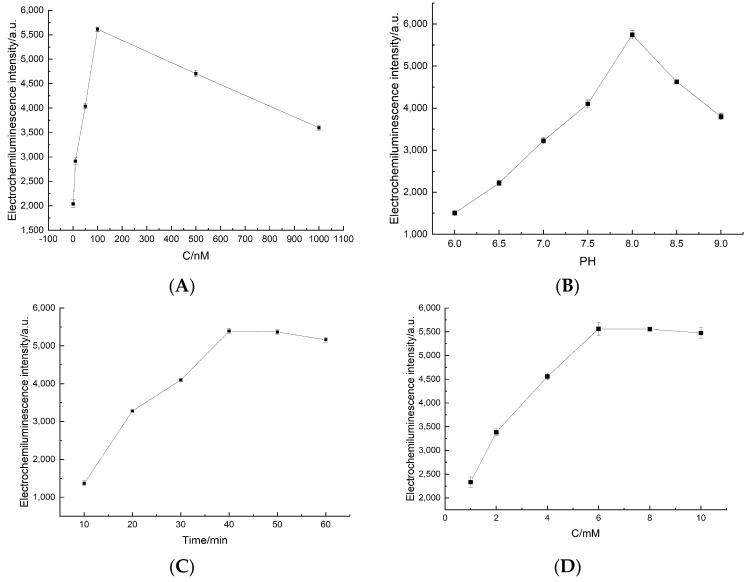
Optimization of the experimental parameters: (**A**) the concentration of the aptamer; (**B**) the effect of the detection solution’s pH; (**C**) the incubation time of pesticides and aptamers; and (**D**) the concentration of H_2_O_2_.

**Figure 6 foods-12-04363-f006:**
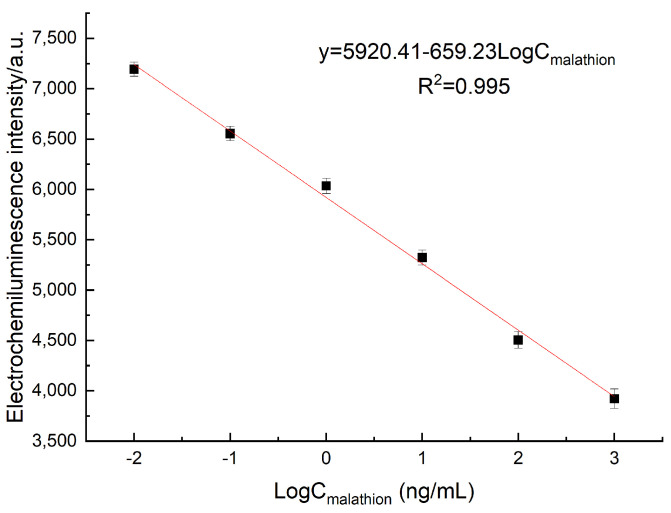
The relationship between ECL intensity and the logarithmic value of malathion concentration.

**Figure 7 foods-12-04363-f007:**
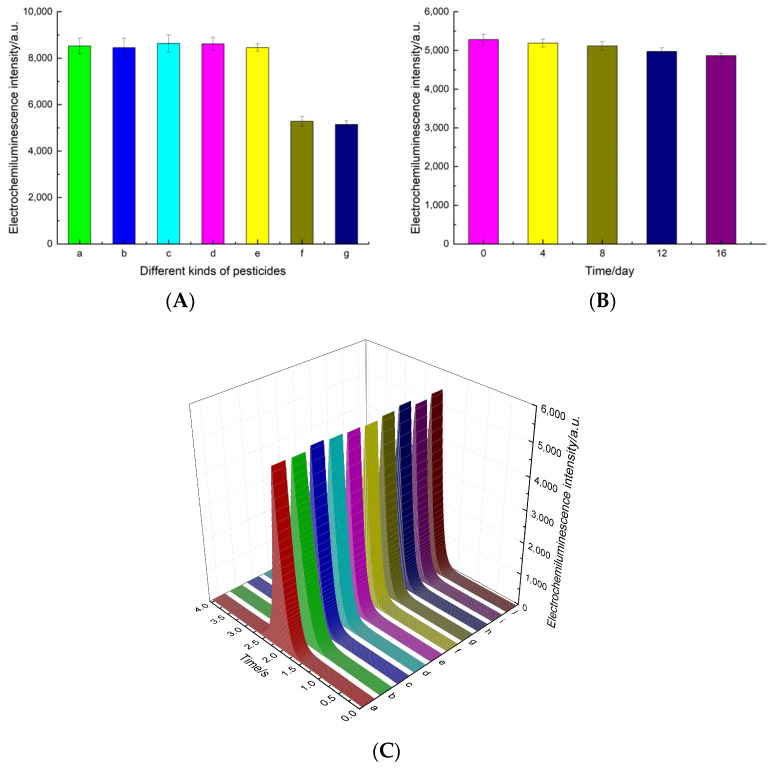
The performance of the aptasensor. (**A**) Selectivity tests of the proposed sensor: (a) phorate, (b) profenofos, (c) acetamiprid, (d) carbofuran, (e) a mixture of the above four pesticides except malathion, (f) malathion, and (g) a mixture of the above five pesticides. (**B**) Long-term stability experiments of the biosensor within 16 days. (**C**) Repeatability inter-assay of the designed ECL aptasensor with five electrodes.

**Table 1 foods-12-04363-t001:** Comparison between proposed aptasensor and other malathion detection methods.

Method	LOD (ng/mL)	Detection Range (ng/mL)	Reference
Gas chromatography	10	10~10^4^	[40]
Colorimetry	60	10~10^5^	[41]
Raman spectra	123	1.23 × 10^2^~1.23 × 10^5^	[42]
Electrochemistry	0.75	3.73~7.47 × 10^4^	[43]
ECL	2.75 × 10^−3^	1.0 × 10^−2^~1.0 × 10^3^	This work

**Table 2 foods-12-04363-t002:** Results of determination of MAL in real samples.

Samples	Addition/(ng/mL)	Detection/(ng/mL)	Recovery (%)	RSD (%)
Cabbage	0	0	-	-
	10	9.421	94.21	1.59
	100	98.337	98.35	1.94
Lettuce	0	0	-	-
	10	9.739	97.39	1.43
	100	99.634	99.63	2.01
Spinach	0	0	-	-
	10	9.945	99.45	1.38
	100	97.482	97.48	2.52

## Data Availability

The datasets generated to obtain the results presented in this article are available from the corresponding author upon reasonable request (xzszhgd@163.com).

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
