# Peer review of "Electrochemiluminescence Sensor Based on CTS-MoS2 and AB@CTS with Functionalized Luminol for Detection of Malathion Pesticide Residues"

_foods, 2023, doi:10.3390/foods12234363_

Round 1

Reviewer 1 Report

Comments and Suggestions for Authors

In this work, Authors reported about the developed an electrochemiluminescence aptasensor for the detection of malathion residues in vegetables. Authors claim, the presented sensor demonstrated lower detection limit and wider detection range. Also was demonstrated detection of malathion pesticide residues in real samples. There are some comments:

1. It is not clear how finally looks the sensor. There are no information for instance of about size of such sensor. It would be nice to show some an optical image of it. Also, I would be recommend to show the surface of sensor - how surface looks, for instance, before functionalization of aptamers. Is it some homogeneous surface or what?

2. It is not clear from the text what kind of MoS2 was used - a powder consisting of flakes or a colloidal solution was purchased. It would be nice to see, for example, the Raman spectra or XPS/EDS analysis whish can show that is MoS2.

3. Fig. 2b It would be nice to show spectra from the fabricated Au NPs

4. The main question. What is the difference between presented manuscript and this publication: Sensors 2015, 15(2), 2629-2643; https://doi.org/10.3390/s150202629? Graphs, diagrams and their sequence look the same. I see the difference only in the tested analytes, but if this is so, the presented work is not the development of a new sensor!

Reviewer 2 Report

Comments and Suggestions for Authors

Mentioned here are the following technical points for improvement:

  • The introduction could be more concise and focused. It should clearly state the research problem, the importance of addressing it, and the research objectives. It would be beneficial to include graphs or figures in the paper to visualize data and illustrate the sensor setup, leading to better comprehension for readers. The chosen keywords are relevant, but it would be beneficial to include specific keywords related to electrochemistry, sensor technology, and pesticide detection for better indexing.
  • The methods section should provide more detailed and step-by-step procedures for the preparation of CTS-MoS2, AB@CTS, AuNPS, PLL-luminol, and the fabrication of the ECL aptasensor. The current description lacks specific details.
  • The results section would benefit from presenting data in a more organized manner, such as tables or graphs. It can make the findings easier to interpret and compare. It's important to include information about the statistical methods used for data analysis, such as the calculation of LOD (Limit of Detection) and RSD (Relative Standard Deviation). This will enhance the scientific rigor of the research.
  • The paper should discuss why certain materials and conditions were chosen. Explain the scientific basis behind the choice of CTS-MoS2, AB@CTS, and other components. This will help readers understand the rationale behind the design of the aptasensor. A more detailed comparative analysis with other methods for pesticide residue detection would be beneficial. This could highlight the advantages and disadvantages of the ECL aptasensor and its potential for real-world applications. Discuss the reproducibility of the results, including the fabrication of the sensor and its performance across multiple trials. Consider investigating the sensor's performance under various environmental conditions. Discuss how the research contributes to the field of sensor technology.
  • Provide insights into future work and potential applications of the ECL sensor technology. This can help guide further research in the field.
  • The conclusion section should NOT succinctly summarize the key findings and their implications.

This feedback should help improve the technical aspects and readability of your research article.

Comments on the Quality of English Language

Moderate editing of English language required

Reviewer 3 Report

Comments and Suggestions for Authors

Authors present the manuscript for the detection of Malathion Pesticide Residues vis ECL techniques. However, this manuscript needs critical major revision before consideration to the next stage.

At present title is too big; please shorten it.

Line 14,” Due to the weak interaction force between MOS2 and AB, they interact through the biocompatibility of  chitos in a chitosan solution” is incorrect for English point of view. What is chitos?

Please avoid the abbreviation from the keywords?

Line 64 need citation, mention other properties and applications of chitosan. You may cite this paper international journal of biological macromolecules 136 (2019) 661-667.

Introduction is too short, no novelty is highlighted, and problem was not identified why this research conducted. Need to rewrite by highlight the above problems.

Overall introduction is very common and all are general information. Need to discuss critically.

Line 74, Check the subscript of MoS2. Check throughout the manuscript.

In the introduction highlighted the method available for the detection. You can cite this paper Polymers 2023, 15(12), 2655 in the manuscript text.

Don’t define always the abbreviation of a same word. for ex: MoS2 is already defined in the introduction section again you define section 2. Check the whole manuscript and avoid it.

Once you define, don’t the full name for ex; chitosan in the introduction as CTS you write chitosan in section 2.3 and so on. you must write CTS. Check for other errors in the whole manuscript.

Line 135: don’t start the sentence with a number. Avoid it.

Section 2 have numerous grammar mistake. Lack of reading flow. Need to rewrite all the sentences critically.

Line 125, make a space number and alphabet for ex, 100mL should be written as 100 mL. check the whole manuscript.

The materials Characterization part is missing in the section. Need to include details.

Section 3.1 has no discussion, scale bar is not visible. Need to provide a histogram graph of particle size.

Include figure legends in each figure.

In table 1 authors mention their work is superior than other why? Need to describe.

LOD is very small in this work any reason with mechanism.

Preliminary materials characterization results are missing. Need to add.

Line 316, it is not novel. Please avoid it.

There is no evidence of CTS-MoS2 surface area data including it.

Need to estimate the catalytic active area from electrochemistry and include in the paper.

Line 329, offered numerous amino groups need to estimate this.

There are no recent references, especially for 2021, 2022 and 2023. Some are here, included it in the manuscript. Inorganic Chemistry Communications 106 (2019) 54-60, Biosensors 2023, 13(3), 348, Microchemical Journal 193 (2023) 109059,

English correction is needed throughout the manuscript.

Improve the reference part.

All figures should be drawn scientifically with software and present high-resolution figures.

The above recommended  references are suggestions and are not mandatory to cite.

Comments on the Quality of English Language

 Extensive editing of English language required

Reviewer 4 Report

Comments and Suggestions for Authors

The development of sensitive detection systems to reveal the presence of pesticides residues in food is of paramount importance in human health protection. The electrochemiluminescence sensor described in your manuscript  can give a contribution in the assessment of organophosphates' contamination, especially in those countries still employing these persistent compounds. The text has been prepared in a good style and language use, nevertheless it must be revised in few but fundamental points.

- Introduction, lines 75-90: this part must be definitely reduced. Please, in this paragraph you must simply summarize the methods employed during the research work.

- Materials and Methods. A very important procedure was missed: Which was the procedure used to spike the real samples? After this, which was the method followed to prepare these samples in order to carry out the electrochemical measurement?

-Results, page 7, line 267: add "concentration" after "malathion". I suggest to use in equation (1) a larger type. 

Page 8: you measured the stability of the sensor's response over a period of 16 days, but the lifespan of the sensor was not mentioned. How long is possible to use it ? Since to prepare the aptasensor requires quite long time this is an interesting parameter to evaluate its suitability in practical applications.

Conclusions. This paragraph should be not a repetition of the paper summary. It must describe the problems that your sensor will solve, the comparison of its performance with known, similar analytical systems, its drawbacks and advantages, as well as possible developments and applications.

Round 2

Reviewer 1 Report

Comments and Suggestions for Authors

Satisfied with the answers

Reviewer 3 Report

Comments and Suggestions for Authors

Authors improved the manuscript.

Comments on the Quality of English Language

Minor checking needed.

Reviewer 4 Report

Comments and Suggestions for Authors

The corrections introduced in the revised text can satisfy the questions I posed on your manuscript and then I can suggest to the Editor that your paper can be accepted and published  in the present version.